# Connecting Western and Eastern Medicine from an Energy Perspective

**DOI:** 10.3390/ijms20061512

**Published:** 2019-03-26

**Authors:** Ming Zhang, Mohamed Moalin, Lily Vervoort, Zheng Wen Li, Wen Bo Wu, Guido Haenen

**Affiliations:** 1Department of Pharmacology and Toxicology, Maastricht University, 6200 MD Maastricht, The Netherlands; z.ming@maastrichtuniversity.nl (M.Z.); m.moalin@maastrichtuniversity.nl (M.M.); l.vervoort@maastrichtuniversity.nl (L.V.); zhengwen.li@maastrichtuniversity.nl (Z.W.L.); w.wu@maastrichtuniversity.nl (W.B.W.); 2Research Centre Material Sciences, Zuyd University of Applied Science, 6400 AN Heerlen, The Netherlands

**Keywords:** western medicine, eastern medicine, opposing forces, redox, Yin–Yang, resilience

## Abstract

Although Western medicine and Eastern medicine are worlds apart, there is a striking overlap in the basic principle of these types of medicine when we look at them from the perspective of energy. In both worlds, opposing forces provide the energy that flows through networks in an organism, which fuels life. In this concept, health is the ability of an organism to maintain the balance between these opposing forces, i.e., homeostasis (West) and harmony (East), which creates resilience. Moreover, strategies used to treat diseases are strikingly alike, namely adjusting the flow of energy by changing the connections in the network. The energy perspective provides a basis to integrate Eastern and Western medicine, and opens new directions for research to get the best of both worlds.

## 1. Introduction

Some people hope that, when their Earthly life is finished, they go to heaven and are spared from burning in hell. What most people do not know is that we do not have to worry about burning in hell; we already burn on Earth. This burning is caused by an unusual suspect, namely oxygen. Most people think we need oxygen and that oxygen is essential for our life. This is correct, but oxygen is also highly reactive and therefore toxic. Because of oxygen, we slowly burn like a candle. This can be exemplified by the development of age pigments. Age pigments are burned proteins that our body cannot remove and therefore accumulate and become toxic [1]. Another example is the burning of DNA that can give rise to cancer [2].

## 2. Opposing Forces

The burning caused by oxygen is called “oxidation”. Luckily, there is also a counter force, namely “reduction”. The first letters of reduction and oxidation combine to give “redox”. Redox describes the equilibrium between the oxidizing and reducing force (Figure 1). As also described in the Introduction, redox energy is the driving force of life; however, redox energy also causes damage. It contains both good and bad.

That everything contains good and bad was coined in the West by Paracelsus in the 15th century who stated that: “A very good thing worth to be acquired, first has to be separated from the bad. The art is such that nothing good can be acquired without a price. To get what you want, you must face also that you do not want” [3]. Within this paradox lies another paradox. An example of this double paradox is that oxidative damage to DNA may, by chance, give rise to a mutated organism that is better adapted to its environment [4]. This is the fundament of the “survival of the fittest” theory of evolution. Thus, “damage” is also essential for survival, resilience and progression.

In the East, it was realized much earlier than in the West that everything contains opposing forces. This is best symbolized by the Tai Ji symbol that visualizes the balance between Yin and Yang. In the black part of the circle, there is a small white circle [5,6]. This visualizes one of the double paradoxes in the symbol. Actually, the Tai Ji symbol is one of the first fractals made by man. The paradoxical nature of oxygen might therefore be better symbolized using the Tai Ji symbol as done in Figure 2, compared to the Western chemical symbol of oxygen.

Yin and Yang are opposing forces that control life, as well as the universe. Yin mainly influences the physical, material aspect and Yang domains the psychological, spiritual aspect of an organism. However, similar to reduction and oxidation, they interact and balance each other. When Yin rises, Yang declines, and when Yin declines, Yang rises [7,8,9]. The opposing forces form the vital energy or Qi that circulates in our body. The organs work together by regulating and preserving Qi through the channels called meridians and collaterals. Diseases will happen when the Qi creates disharmony. Treatment aims to correct the Yin–Yang balance, thus Qi is readjusted and harmony is restored [10].

## 3. Western Concept of Health

In our life, the redox energy changes (Figure 3). In a healthy body, this redox energy is high and kept within very strict limits in our body. We call this homeostasis, a harmony comparable to the Yin–Yang balance. Health can be defined as the ability to adapt to challenges [11]. When we have stress, a challenge, there is an attack on our energy. The energy goes down, but our body has resilience to absorb the challenge and bounce back. In the rebound, there is even a temporary improvement and you build up resistance [12,13,14,15]. Thus, a little bit of bad is good for you, which is in line with the double paradox described above. We do have a problem when the challenge is intense and prolonged. For example, due to a chronic illness, the energy decreases and reaches a new equilibrium at a lower energy level [16]. In this condition, we can no longer cope with even a small challenge. All energy might flow away and then it is the end.

Where does this resilience of our body come from? In Western science, the body is made up of molecules, and all molecules are closely linked together. These connections can be visualized by the network plots that are generated in system biology (Figure 4) [17,18]. A molecule can be converted into another molecule, in a process often catalyzed by enzymes and controlled by various other molecules. The energy for these connections comes from redox. Together, these connections form an elastic safety net that easily can absorb a challenge and even can give a rebound that increases resilience. In a sick organism, many connections are lost, and the safety net becomes fragile. This will reduce resilience [19]. The importance of connections is easily remembered with an elegant mnemonic. The first two letters of “wellness” give “we”. When “we” are together, “we” are connected and healthy. This is true in every society and culture. The first letter of “illness” gives “i”. When “I” am alone, “I” am isolated and this is not healthy [20].

## 4. Eastern Concept of Health

Connections are also very important for resilience and health in the East. In Chinese philosophy, all organisms are made up of the five symbolic elements: wood, fire, earth, metal and water (Figure 5). These five elements interact and compensate each other to find the proper balance. An element can generate another element in the generation cycle. In the destruction cycle, an element restrains another element [21]. The interaction between the elements is governed by the Yin and Yang balance. Each organism is constantly striving for an optimal dynamic balance or harmony of these five elements, which is associated with a healthy condition. In this condition, all vital organs of the body can cope with challenges, which gives resilience. An imbalance between the elements in this network will be manifested as a disease.

## 5. Connecting East and West

Despite the big differences between Eastern and Western medicine, apparently, the above shows that the basic principle used are strikingly similar. Both make use of opposing forces that create energy that flows through a network to fuel life. As outlined in the next paragraph, both types of medicine make use of this concept to treat diseases or increase health. To increase health, both aim to strengthen connections or make new ones to readjust the balance, i.e., homeostasis (West) or harmony (East), and strengthen the network (resilience).

### 5.1. Western Medicine

In Western science, connections are made when molecules react with each other and this can offer protection [22,23,24]. An important question is why some connections are made while others are not; why reactions between some molecules happen and others do not; and how the energy flows through the network. It appears that this can be explained according to Pearson’s hard–soft–acid–base concept [25,26]. In the hard–soft–acid–base concept, “hard” applies to species which are relatively small, have a high charge and are weakly polarizable, and “soft” applies to species which are relatively big, have low or no charge and are strongly polarizable. Consequently, “hard” acids (electrophiles) and “hard” bases (nucleophiles) have a large orbital overlap and therefore react fast with each other. In addition, due to the large orbital overlap, “soft” electrophiles react fast with “soft” nucleophiles. The hard–soft–acid–base concept can also be generalized for other types of electrophiles and nucleophiles that are not necessarily an acid or a base [27].

When the energy is concentrated in one place of a molecule, similar to the focal point of a lens, this molecule becomes “hard” and very reactive. For example, in the hydroxyl radical, all the energy of the unpaired electron is located on a single atom (i.e., oxygen). It is therefore the most reactive molecule in our body (Figure 6). The hydroxyl radical reacts quickly with virtually any molecule it encounters. This will cause an uncontrolled energy flow and inflict damage to our body. Antioxidants such as quercetin and 7-mono-*O*-(β-hydroxyethyl)-rutoside (monoHER) are hard electron donors that react quickly with the hydroxyl radical and other reactive oxygen species (ROS) that are hard electron acceptors. By accepting electrons, ROS are converted into relatively unreactive compounds, that do not damage the cell. For example, by accepting an electron the hydroxyl radical is converted into the hydroxyl anion that, after protonation, becomes water. By donating electrons, quercetin and monoHER are converted into their corresponding soft oxidation products, quercetin quinone methide and monoHER quinone, respectively (Figure 6). These oxidation products act as “soft” electrophiles and selectively react with “soft” nucleophiles such as thiols instead of “hard” nucleophiles such as water. As outlined in the next paragraph, we can explain the biological effect of a challenge and the modulating effect by antioxidants using this hard–soft–acid–base concept.

A challenge on the redox energy often comes through hard energy of ROS. The hard energy of ROS will damage the cells. This is because the hard energy can react with everything in the cell including proteins, lipids and DNA, and this damage will eventually destroy the cell. There is also a sensor in the cell that is switched on by the hard energy. By switching on the sensor, the cell reacts and makes more antioxidants. This sensor is Kelch-like ECH-associated protein 1 (KEAP1). ROS can oxidize the thiol groups on KEAP1, and the oxidized KEAP1 will dissociate from its complex with Nuclear factor erythroid 2-related factor 2 (Nrf2). The freed Nrf2 will translocate to the nucleus and stimulate the transcription of numerous detoxifying and antioxidant genes (Figure 7A). A small challenge will eventually strengthen the defense system and increase the resilience of the cell [28,29].

When we add an antioxidant such as monoHER, the hard energy of ROS is absorbed and softened by the formation of the soft monoHER quinone (Figure 7B). The soft energy of this quinone does not damage DNA and other vital cell molecules, but it can turn on the sensor very well. This is because the switch on the sensor is a soft nucleophile, namely a thiol or sulfhydryl (SH) group. In line with the hard–soft–acid–base concept, a soft quinone will react efficiently and selectively with the soft thiol group on KEAP1 [12,29]. The conversion of hard into soft energy explains the direct protection as well as the efficient increase in resilience given by antioxidants. Thus, the antioxidant does not only give protection against a challenge; it also increases the rebound due to a challenge. This gives more resilience.

### 5.2. Eastern Medicine

In Eastern philosophy, when there is an imbalance between Yin and Yang, the flow of energy is incorrect and harmony is lost, which is manifested as a disease. The imbalance of energy can be corrected by acupuncture or moxibustion of meridian sites, or by Traditional Chinese Medicine (TCM). These treatments aim to readjust the Yin–Yang energy to regain the harmony of five elements.

TCM mostly consists of a combination of medicinal herbs. Their combination is based on the rule of “Jun-Chen-Zuo-Shi”, known as the Four Responsible Roles (Figure 8). Each herbal ingredient has a specific function, and their interplay makes the therapy more powerful and specific. The “Jun” (emperor) herb is the principle active herb, having the main effect in treating the disease, similar to a hard molecule. The “Chen” (minister) herb strengthens the curative effect of the Jun herb. The “Zuo” (adjuvant) herb modulates the effects of Jun and Chen and the “Shi” (messenger) herb harmonizes the effect of the other ingredients, similar to how the soft antioxidant channels the hard energy. Thus, by combining the herbs, the main therapeutic effect is enhanced, and the side effects are reduced [30,31].

For instance, in the formula Ma Huang Decoction developed to treat, e.g., asthma (Figure 9), Ma Huang is the most effective herb and is the Jun herb. It is used to promote the sweating, and dilatate the lung. Gui Zhi serves as the Chen herb to assist Ma Huang, but it induces sweating and reduces the side effects of the Ma Huang by warming the channels to reduce headache and pantalgia. Xing Ren and Gan Cao work together as the Zuo and Shi herbs, respectively, to improve the therapeutic effects of Ma Huang and Gui Zhi. Xing Ren, which is the Zuo herb with a bitter and warm nature, increases the lung function and strengthens the sweating effect of Ma Huang. Gan Cao, the Shi herb with sweet and warm nature, harmonizes the action of the other herbs and makes the therapeutic effect of the combination more specific. Through combing herbs, the therapeutic effect of Ma Huang is enhanced, and the side effects caused by Ma Huang are also reduced to a relative safe level. The intricate interaction between compounds in TCM harmonies the heat and the energy in the body to attain good health [32,33].

## 6. Conclusions

Western and Eastern philosophies of health seem to be worlds apart. In the West, an organism is reduced to a collection of molecules. This approach has been quite successful in generating effective medicines. However, the limitations of Western medicine are becoming more evident. Apparently, in the Western reductionistic approach, something is lost. This is probably best evidenced by “emotion” and “feeling” that—although essential in life—cannot be explained only by looking at molecules. Eastern medicine uses a more intuitive approach. Although, from a Western perspective, Eastern medicine is not “scientific”, people have profited from this intuitive type of medicine for thousands of years and Eastern and other traditional types of medicine are still used by the majority of the people. Despite the difference, both worlds appear to be based on a similar principle when we look at Western and Eastern medicine from the perspective of energy (Figure 9). The universal principle is that opposing forces generate the energy that flows through networks in an organism, which fuels life. This perspective might provide a basis to integrate Eastern medicine and Western medicine and opens new directions for research to fill the knowledge gaps between both worlds.

## 7. Future Perspective

The energy perspective creates a bridge to connect Eastern and Western medicine. In the West, we might benefit more from the dynamic interaction between molecules, one of the fundaments of TCM. Using isolated arteries, we could confirm that the accompanying herbs in the Ma Huang Decotion can mitigate the side effects of Ma Huang [33]. Moreover, the dynamic interaction between several herbs in TCM on muscarinic receptor binding are also—from a Western point of view—unexpected and even contra-intuitive [34]. The energy perspective also indicates that we still need to extend our knowledge on how antioxidants that differ in hardness/softness will have different redox modulation effects [35]. Another opportunity is to study other forms of energy, e.g., light. An interesting finding is that TCM “corrects” the light transmitted by the body [36]. This cannot be explained with the Western reductionistic approach, yet. There are numerous other mysterious “forces” in Eastern medicine and other types of traditional medicine that lack a “Western scientific basis”, and therefore are left unused and might even be lost. Undoubtedly, medicine will improve when East and West become more connected, and we get the best of both worlds.

## Figures and Tables

**Figure 1 ijms-20-01512-f001:**
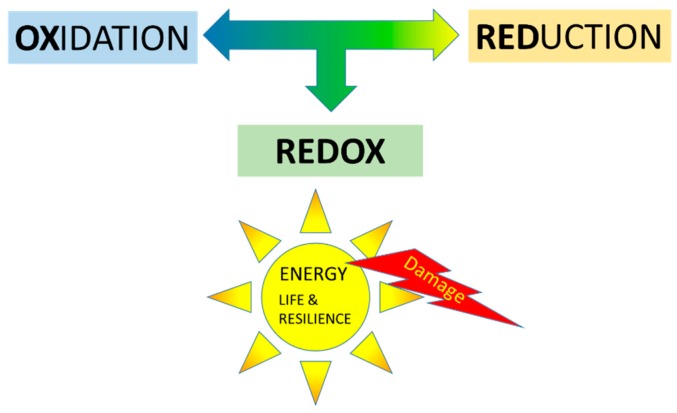
Redox, the equilibrium between oxidation and reduction, is the energy source for life, as well as damage.

**Figure 2 ijms-20-01512-f002:**
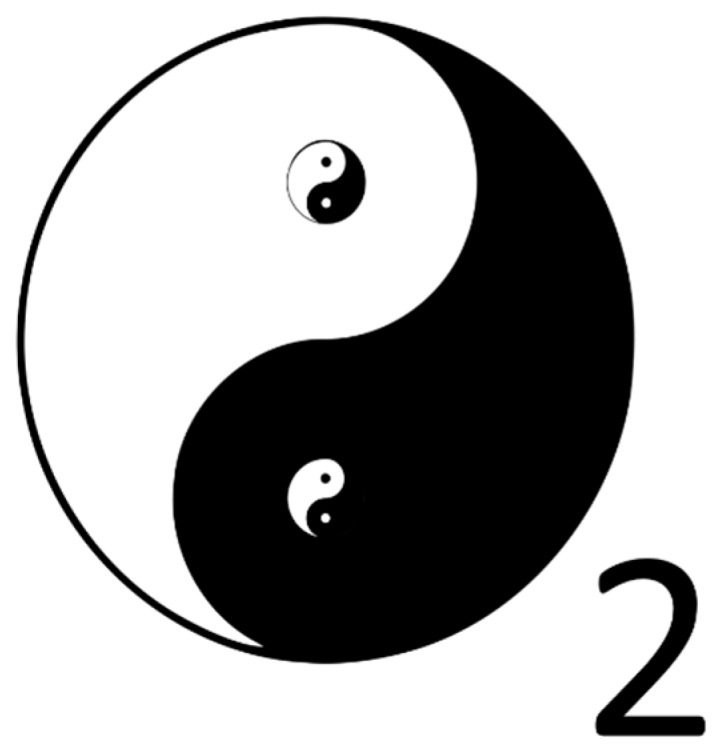
The symbol that might visualize the nature of oxygen more accurately than the Western chemical symbol (O_2_). Oxygen is essential for life but it is also reactive and therefore toxic. This dual nature is resembled in the Tai Ji symbol.

**Figure 3 ijms-20-01512-f003:**
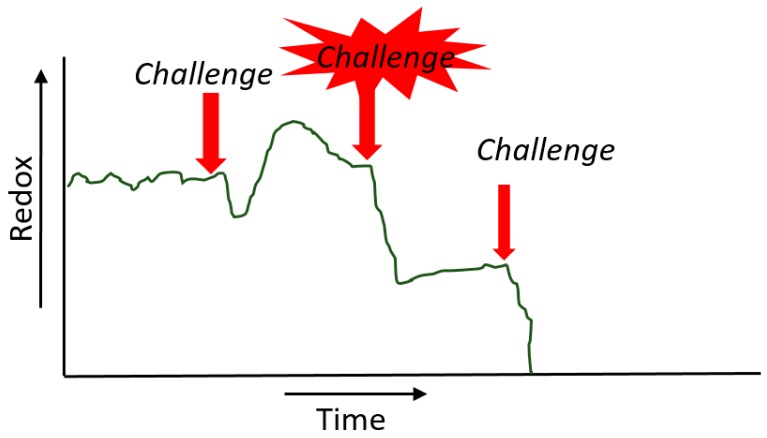
Change of the redox energy in our life. The explanation is given in the text.

**Figure 4 ijms-20-01512-f004:**
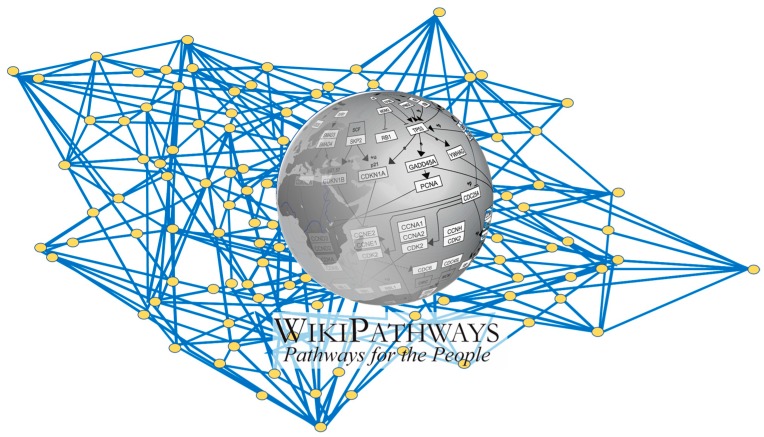
The network formed by biochemical reactions. The background gives a schematic impression of how molecules (yellow dots) in a cell are connected by chemical reactions (blue lines), illustrating that together they form an “elastic safety net” that can deal with challenges. An overview of these reactions is given by the interactive “WikiPathways” platform (Available online: https://en.wikipedia.org/wiki/WikiPathways) an open community resource dedicated to collect all interactions among molecules in a cell.

**Figure 5 ijms-20-01512-f005:**
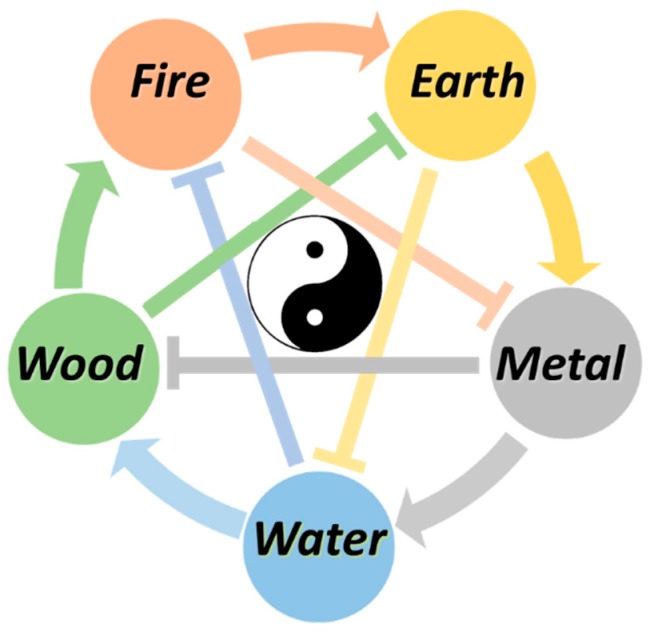
Basic principle of the five elements theory. Each element is generated by another (generation cycle), but also controlled by another one (destruction cycle). Wood generates fire, fire generates earth, earth generates metal, metal generates water and water generates wood, as depicted by the colored curved arrows. Meanwhile, wood controls earth, earth controls water, water controls fire, fire controls metal and metal controls wood, as depicted by the colored T arrows. The interaction between the elements in this network is governed by Yin and Yang, symbolized by the Tai Ji symbol.

**Figure 6 ijms-20-01512-f006:**
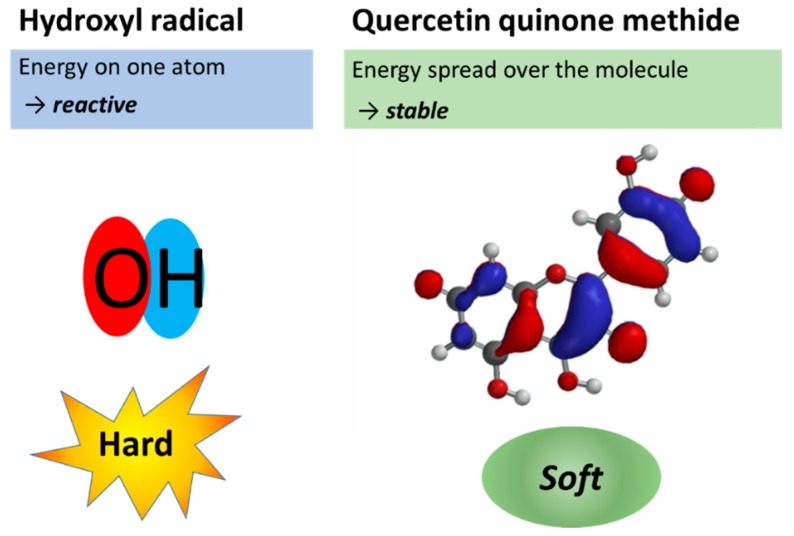
Singly occupied molecular orbital of the hydroxyl radical and lowest occupied molecular orbital map of quercetin quinone methide calculated with Spartan’18. In the hydroxyl radical, all energy (in red) is concentrated on the oxygen atom, making it a very reactive, hard molecule. In the quercetin quinone methide molecule, the energy is spread over the molecule, thus it is a relatively unreactive, soft molecule.

**Figure 7 ijms-20-01512-f007:**
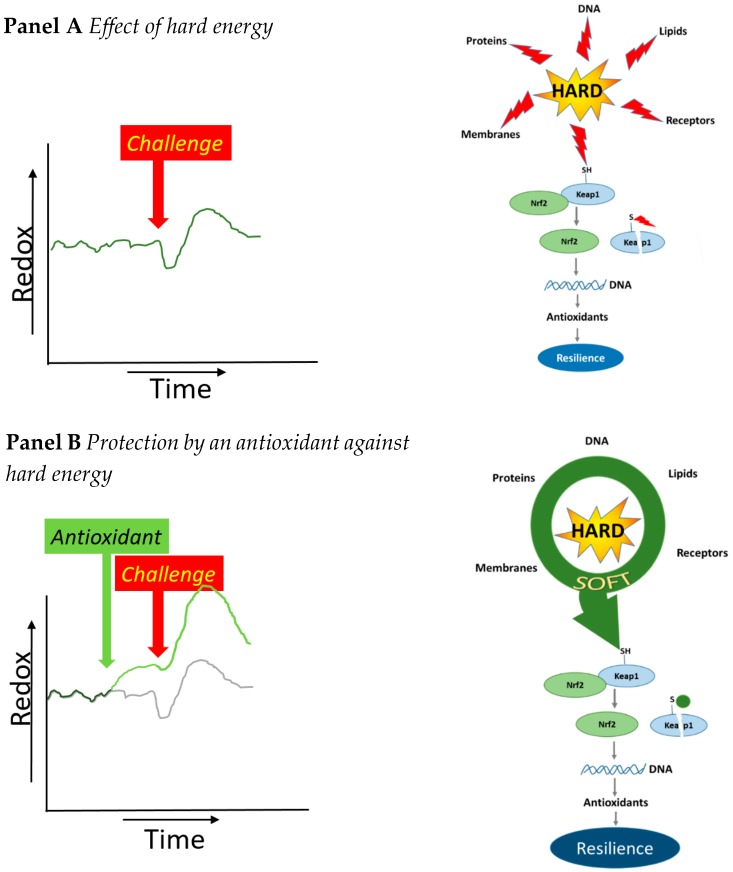
The biological explanation for the difference in the effect of hard (**A**) and soft (**B**) energy on the redox energy. In the left figure of panel A, the dark green lines shows the change of redox energy due to a challenge. In the left figure of panel B, the green line shows how an antioxidant increases the redox energy. In the presence of an antioxidant, the damage by the challenge is less, and the rebound is higher than without antioxidant (grey line). The right figures in both panel A and panel B give the biological explanation. A “challenge” often comes from hard energy. The hard energy damages every molecule in the cell and is non-selective and inefficient in reacting with the thiol (SH) group on Kelch-like ECH-associated protein 1 (KEAP1), that switches on the Nuclear factor erythroid 2-related factor 2 (Nrf2) pathway. An antioxidant can convert the hard energy into soft energy that gives little damage, but the soft energy is relatively efficient to react with KEAP1, switching on the Nrf2 pathway. This is because the soft energy reacts efficiently and selectively with the soft thiol (SH) group on KEAP1.

**Figure 8 ijms-20-01512-f008:**
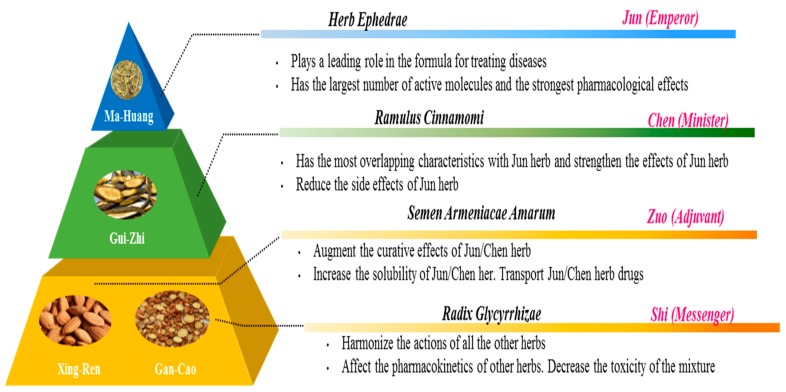
diagram of the combination principle of Traditional Chinese Medicine formula, adapted from Yao et al. (2013) [30].

**Figure 9 ijms-20-01512-f009:**
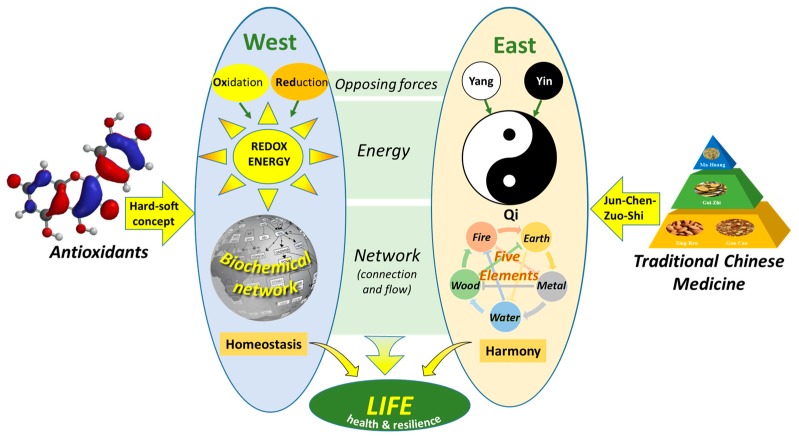
The connection of Western medicine and Eastern medicine from energy perspective. In both worlds, opposing forces generate the energy that flows through networks, which fuels life. Antioxidants interact with other molecules based on the hard–soft–acid–base concept which can be used to regain homeostasis. In TCM, different herbs are combined based on the rule of “Jun-Chen-Zuo-Shi” to restore the energy of Qi in the network to regain harmony.

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
