# Peer review of "Connecting Western and Eastern Medicine from an Energy Perspective"

_ijms, 2019, doi:10.3390/ijms20061512_

Round 1
Reviewer 1 Report
The goal of this review paper is to determine potential connecting between the western and eastern medicine from an energy perspective.
To gain this goal authors’ used available literature search to gain their goal. Because although Western medicine and Eastern medicine are worlds apart, there is a striking
overlap in the basic principle of these types of medicine when we look at them from the perspective of energy. In both worlds, opposing forces provide the energy that flows through networks in an organism, which fuels life. In this concept, health is the ability of an organism to maintain the balance between these opposing forces, i.e. homeostasis (West) and harmony (East), which creates resilience. Moreover, strategies used to treat diseases are strikingly alike, namely adjusting the flow of energy by changing the connections in the network. Authors’ conclusion is that the energy perspective provides a basis to integrate Eastern and Western medicine, and opens new directions for research to get the best of both worlds.
I am extremely impressed with authors’ idea and concepts.
If I am correct this is one of first critically analyzed report regarding western and eastern medicine especially regarding the diagnosis and treatment strategies from the energy stand point.
Manuscript well written and illustration completely support authors’ working hypothesis..
However at the present form this manuscript contain some of stylistic and typo errors and therefore requires proof reading by the person with native English.
Author Response
We would like to thank reviewer 1 for the time spend in reviewing our manuscript and for the positive comments. The reviewer indicated that our manuscript contained some errors. We have corrected these mistakes to the best of our ability. Also based on the remark and suggestions form the other reviewer, we revised some of the text which makes our message more clear. The changes are made visible with the “track changes” option used in revising the manuscript.
Reviewer 2 Report
In my opinion, the manuscript sounds like it was written for a popular science magazine. I certainly would not call it a scientific review. It is more like a set of facts, statements and some philosophical considerations provided to prove the validity of the created hypothesis.
I think that the subject is interesting and I do not claim that the authors are wrong. I agree that Western medicine would benefit from the achievements of the East. However, I do not feel that the article can somehow bring the two worlds closer. I am not convinced by the examples and observations presented. Probably because in my opinion some issues are treated too generally (e.g. L. 36-38; L. 69-76; L. 79-86; comparison between antioxidants and Jun-Chen-Zuo-Shi (Fig.9); what about the application of physics, psychology and psychotherapy in Western medicine?).
Some other issues:
L. 43-45 – Organism itself do not learn from the mistakes and errors of DNA. It is rather that the environment verifies whether a change in DNA is beneficial or not.
Figure 3. Has the change of redox energy in human life ever been studied?
L. 228-230 – Sentence is not clear.
Author Response
Reply to reviewer 2:
We would also like to thank reviewer 2 for her time spent in reviewing our manuscript and her constructive remarks. The reviewer identified some parts of the manuscript that she finds too general. We have carefully reviewed these parts and would like to answer with the following remarks and/or changes.
Remark 1. L. 36-38
That redox contains both good and bad was already covered in the introduction part. This was also mentioned in line 36-38. We now connect both by referring in the text of line 36-38 to introduction. We think this is better than repeating the information given in the introduction.
Remark 2. L. 69-76;
We have added more references to further substantiate our content. Please also see the comments on figure 3 (remark 7) that deals with the same issue.
Remark 3. L. 79-86
In this paragraph, the concept of resilience is introduced. This is extensively elaborated in line 122-182 and figure 7. Therefore, we think it is not necessary to elaborate this in line 79-86.
Remark 4. Comparison between antioxidants and Jun-Chen-Zuo-Shi (Fig.9)
Figure 9 shows the overlap between Western and Eastern medicine. The figure depicts that in both worlds, opposing forces generate the energy that flows through networks, which fuels life. Antioxidants interact with other molecules based on the hard-soft-acid-base concept which can be used to regain homeostasis. In TCM, different herbs are combined based on the rule of Jun-Chen-Zuo-Shi to restore the energy of Qi in the network to regain harmony.
We have added this explanation to the legend of figure 9. We hope the reviewer now better understands how we compare antioxidants with TCM, and how we compare the hard-soft-base-acid with Jun-Chen-Zuo-Shi. We have also revised line 218 -223 to present the concept more clearly.
Remark 5. What about the application of physics, psychology and psychotherapy in Western medicine?
We did not include psychology therapy and alternative medicine that is used in the West. Actually, this is an example of a double paradox, namely that in the West “unscientific” and non-evidence based therapies are also considered to be useful. Including this double paradox would make the review too complicated. Actually, the use of “alternative” medicine in the West support our message that the West will benefit from other types (Eastern) of medicine
Remark 6. L. 43-45 – Organism itself do not learn from the mistakes and errors of DNA. It is rather that the environment verifies whether a change in DNA is beneficial or not.
The comment of the reviewer indicates that the metaphor we used is not clear, therefore, we revised the text, deleting the metaphor and more clearly stating what we mean. Changes are made in line 44-45.
“An example of this double paradox is that oxidative damage to DNA may, by chance, give rise to a mutated organism that better fits in its ever-changing environment. This is the fundament of the “survival of the fittest” theory of evolution. So “damage” is also essential for survival, resilience and progression. “
Remark7. Figure 3. Has the change of redox energy in human life ever been studied?
The actual change of redox energy during life has not been studied. Homeostasis will keep this within strict limits. It is well documented that during an oxidative challenge, the redox status measured e.g. the ratio of GSH/GSSG will drop. Moreover, the rebound due to the KEAP1 pathway is also well documented. This is also known as hormesis. Moreover, it is known that in chronic diseases the redox status is reduced. In the legend of the figure 3, we refer to the text on this subject (Line 69-76). We have now better substantiated by incorporating more references (This remarks also relates to the second remark of the reviewer).
Lemmens, K.J.; Sthijns, M.M.; van der Vijgh, W.J.; Bast, A.; Haenen, G.R. The antioxidant flavonoid monoHER provides efficient protection and induces the innate Nrf2 mediated adaptation in endothelial cells subjected to oxidative stress. PharmaNutrition 2014, 2, 69-74.
Cook, R.; Calabrese, E.J. The importance of hormesis to public health. Environmental Health Perspectives 2006, 114, 1631-1635.
Bravi, M.C.; Armiento, A.; Laurenti, O.; Cassone-Faldetta, M.; De Luca, O.; Moretti, A.; De Mattia, G. Insulin decreases intracellular oxidative stress in patients with type 2 diabetes mellitus. Metabolism 2006, 55, 691-695.
Remark 8. L. 228-230 – Sentence is not clear.
We agree with the reviewer, and we have revised the paragraph on the future perspective to present this more clearly (Line 223-230).
“The energy perspective creates a bridge to connect Eastern and Western medicine. In the West, we might benefit more from the dynamic interaction between molecules, one of the fundaments of TCM. Using isolated arteries, we could confirm that the accompanying herbs in the Ma Huang Decotion can mitigate the side effects of Ma Huang. Moreover, we found that the dynamic interaction between several herbs in TCM on muscarinic receptor binding also - from a Western point of view - unexpected and even contra intuitive.
The energy perspective also indicates that we still need to extend our knowledge on how antioxidants that differ in hard/softness will have different redox modulation effects. The energy perspective also inspires to examine the impact of other forms of energy, e.g. light. An interesting finding is that TCM "corrects" the light transmitted by the body. This cannot be explained with the Western reductionistic approach, yet. There are numerous other mysterious "forces" in Eastern medicine and other types of traditional medicine that lack a "Western scientific basis", and therefore are left unused and might even be lost. Undoubtedly, medicine will improve when East and West become more connected, and we get the best of both worlds.”
Round 2
Reviewer 2 Report
I did not change my opinion about this manuscript. From my point of view, contemporary Western medicine is multi-level and we cannot reduce it only to the influencing the redox reactions. I cannot agree that psychology and psychotherapy used in modern medicine are “unscientific” or “non-evidence based therapies”. The more we cannot omit physical phenomena which are applied in medical diagnosis and therapy methods e.g. in lasers, radiotherapy, fototherapy, magnetotherapy, magnetostimulation etc. Authors claim that they do not want to introduce some knowledge because it will make their paper “too complicated". In my opinion, without a wide and in-depth look, the paper remains superficial.
Author Response
We would like to thank reviewer 2 again for her time spend in reviewing our manuscript and expressing her challenging opinion. Apparently, our revision and rebuttal did not convince her. Therefore, we again had an in-depth look on how we convey the subject. In our previous rebuttal we provocatively stated that the fundaments of Western therapies are not as solid as we might think (see e.g. http://www.na-businesspress.com/JLAE/FriedmanHH_Web14_2_.pdf ), and are made of postulates, or even of paradigms, presumptions and prejudices, and perhaps ultimately even of hubris. We would welcome it when the reviewer would direct us to the ultimate “scientific” fundaments of e.g. psychotherapy and magnetotherapy. In our previous rebuttal we argued that the ultimately undefinable “in-depth” origin of “hard” Western science actually underlines our message. The reader of our manuscript herself/himself has to decide whether she/he agrees with our conclusion (i.e. that the fundament/origin of Western and Eastern science in essence is quite similar). Apparently, this was the case with reviewer 1, and we were pleased that the editor honestly stated that he is puzzled by the subject (just as we still are). We hope that after this explanation we now have succeeded in getting at least the benefit of the doubt of reviewer 2. We await her response.
Round 3
Reviewer 2 Report
I have no further comments.